# Detection and Characterization of Active Slope Deformations with Sentinel-1 InSAR Analyses in the Southwest Area of Shanxi, China

**Xuguo Shi [1], Li Zhang [2,\*], Yulong Zhong [1], Lu Zhang [2,\*] and Mingsheng Liao [2]**

[1]  School of Geography and Information Engineering, China University of Geosciences (Wuhan), 68 Jincheng Street, Wuhan 430078, China

[2]  State Key Laboratory of Information Engineering in Surveying, Mapping and Remote Sensing, Wuhan University, Wuhan 430079, China

\*  Correspondence: lizhang_ylln@whu.edu.cn (L.Z.); luzhang@whu.edu.cn (L.Z.); Tel.: +86-27-6877-8395 (L.Z. & L.Z.); Fax: +86-27-6877-8229 (L.Z. & L.Z.)

**Abstract:** A catastrophic landslide happened on 15 March 2019 in Xiangning County of Shanxi Province, causing 20 fatalities. Such an event makes us realize the significance of loess slope instability detection. Therefore, it is essential to identify the potential active landslides, monitor their displacements, and sort out dominant controlling factors. Synthetic Aperture Radar (SAR) Interferometry (InSAR) has been recognized as an effective tool for geological hazard mapping with wide coverage and high precision. In this study, the time series InSAR analysis method was applied to map the unstable areas in Xiangning County, as well as surrounding areas from C-band Sentinel-1 datasets acquired from March 2017 to 2019. A total number of 597 unstable sites covering 41.7 km$^2$ were identified, among which approximately 70% are located in the mountainous areas which are prone to landslides. In particular, the freezing and thawing cycles might be the primary triggering factor for the failure of the Xiangning landslide. Furthermore, the nonlinear displacements of the active loess slopes within this region were found to be correlated significantly with precipitation. Therefore, a climate-driven displacement model was employed to explore the quantitative relationship between rainfall and nonlinear displacements.

**Keywords:** Sentinel-1 SAR Interferometry; loess; precipitation; landslide; coal mining

## 1. Introduction

Loess is an aeolian deposit that was accumulated over the past 2.5 million years in arid and semi-arid climate zones [1]. China has the widest, most diversified and thickest loess accumulation area in the world, called the Loess Plateau covering an area of 640,000 km$^2$. Usually loess slope can remain stable for a long time under arid conditions. However, it is prone to collapse with the increasing water content [2,3]. Landslides are the most common geological disasters in the Loess Plateau in northwestern China. According to a recent study, 14,544 loess landslides are distributed in the Loess Plateau and the loess slope stabilities are affected by multiple factors such as soil properties, geomorphic structure, and rainfall [4]. In recent decades, changes in the natural environment together with intensive anthropogenic activities have caused frequent landslides. Loess landslides posed great threats to the safety of people's lives and properties. The rainfall induced Bailu tableland landslide occurred at 17 September 2011 and brought 32 casualties [5]. The seasonal irrigations in the Heifangtai terrace and the South Jingyang terrace make the ground water level rapidly increased and triggered retrogressive landslides [6–11]. As a new example, a slope

collapsed in Xiangning County, Shanxi Province on 15 March 2019, and more than 20 people were killed by this disaster.

Identification of active loess landslides is an essential task for landslide prevention and disaster early warning. Numerous techniques have been employed to measure the displacement of slopes such as global positioning systems (GPS), crack gauges [10], Synthetic Aperture Radar (SAR) Interferometry (InSAR) [2,10,12], light detection and ranging (LiDAR) [13] and ground-based InSAR [14]. In recognition of the unique advantages of wide-coverage and high sensitivity of measuring displacements, satellite SAR images have been more and more widely used in landslide identification and monitoring. In general, the amplitude information can be employed to extract deformation of landslide at submeter level [15–18] while the phase information can be used to monitor landslides with deformation from millimeter level to tens of centimeters [10,19]. Along with the accumulation of SAR data and development of time series InSAR analysis methods such as persistent scatterer InSAR (PSI) [19,20], small baselines subset InSAR (SBAS) [21,22], and SqueeSAR [23], InSAR technology has been gaining acceptance as a popular and powerful geodetic tool by the scientific community.

However, the InSAR technique is rarely employed in geohazards identification and monitoring in the Loess Plateau [24]. It has been successfully used in landslides detection in terrace areas (e.g., the South Jingyang [12] and Heifangtai terrace [10,11,25]) with high resolution TerraSAR-X datasets and deformation monitoring due to urban expansion and mountain evacuation (e.g., Lanzhou [26] and Yan'an [27]) with moderate resolution Environmental Satellite (Envisat) Advanced Synthetic Aperture Radar (ASAR) and Sentinel-1 datasets. Wide-area unstable loess slope identification in the upstream of Yellow River [2] or tectonic displacements in the Linfen-Yuncheng basin [28] have also been carried out with multiple InSAR datasets. The obtained results agreed well with previous investigation or ground measurements. Meanwhile, irrigation or rainfall induced loess slope displacements are observed from InSAR measurements [2,10]. However, the quantitative relationship between displacements and impact factors, e.g., precipitation or ground water level changes are not described in these studies.

The primary objective of this study is to identify the active loess landslides in the southwest part of Shanxi Province (including Xiangning and surrounding counties) using Sentinel-1 SAR observations from March 2017 to 2019. The characteristics of active landslides are analyzed from time series InSAR results. The inferred triggering factors of the Xiangning landslide are also given. Furthermore, the distribution of landslides, as well as the relationship between displacement and rainfall are analyzed. This paper is organized as follows. Section 2 introduces the study area and datasets. Section 3 describes the workflow of our method. Sections 4 and 5 present the results and discussions. The conclusions are given in the end.

## 2. Study Area and Datasets

### 2.1. Study Area

The Loess Plateau is the most concentrated and largest area of loess accumulation on the earth [2,29]. It stretches over seven provinces of north China. The peculiar lithological composition is highly prone to landslides and other geological hazards due to the intensive anthropogenic activities (e.g., construction and irrigation) and natural environmental factors (e.g., rainfall and tectonic movements) [2].

Our study area is located on the border between Shanxi and Shaanxi Provinces, covering almost five counties including Xiangning County, Ji County, Hancheng City, Hejin City, and Jishan County to the northwest of Linfen Basin and Yuncheng Basin, as shown in Figure 1a. The catastrophic failure of the Xiangning landslide that stimulated this study is marked with a black star in Figure 1a. It is located at the Zaoling town, southwest of Xiangning County. The middle reach of the Yellow River passes through our study area from north to south. In addition, the second largest tributary of the Yellow River, the Fen River, also flows through our study area.

The terrain of our study area is characterized by hillslopes with an altitude ranging between 300 and 2000 m a.s.l. [30]. The surface is covered with a large amount of well-developed Quaternary unconsolidated loess. Erosion is extremely prevalent widely, which provides favorable conditions for geological disasters. In addition, coal resources are widely distributed. As a result, mining activities can easily induce disasters such as subsidence and landslide in mountainous areas [31]. A normalized differential vegetation index (NDVI) map produced from Sentinel-2 acquired on 5 June 2018 of our study area is given in Figure 1b. There are dense vegetations in the east and west side of the mountainous region indicated by high NDVI values close to 0.8. However, the vegetation coverage in Xiangning and the surrounding areas is sparse which is favorable for InSAR analysis.

The study area is a typical temperate continental monsoon climate with dry and windy springs, hot and rainy summers (high rainfall intensity and concentration), cool and humid autumns, and cold winters. The annual rainfall is 480–600 mm. The freezing period and the depth of frozen soil vary with the terrain and location. The general annual freezing period is from October to March.

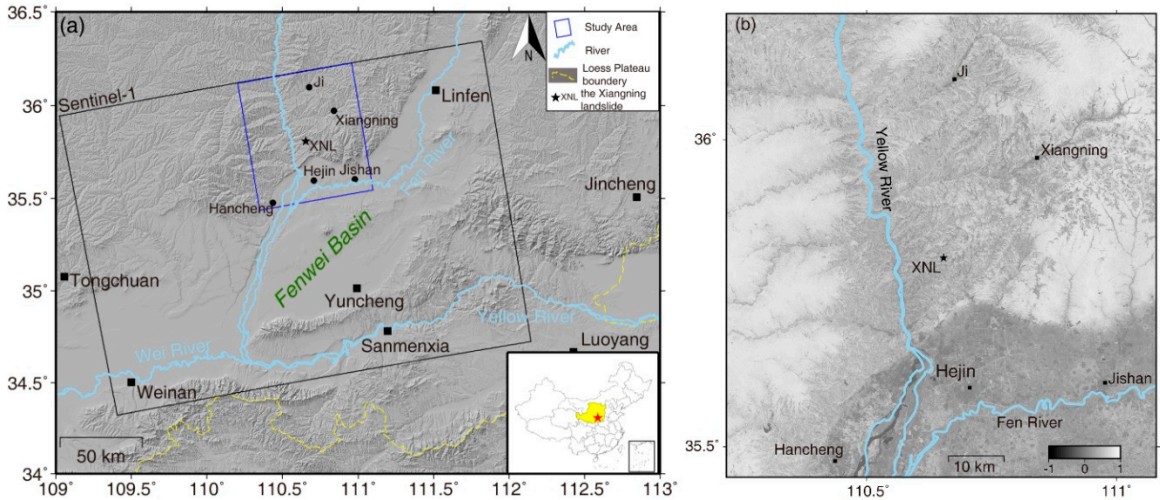

**Figure 1.** (**a**) Location of our study area. The yellow polygon and the red star in the inset represent the Loess Plateau and the location of our study area. (**b**) The normalized differential vegetation index (NDVI) map created from Sentinel-2 acquired on 5 June 2018.

### 2.2. Datasets

Fifty-nine Sentinel-1 SAR images acquired in IW mode from March 2017 to 2019 in ascending orbit were collected. The coverage of the datasets is shown in Figure 1 marked with the black rectangle. The interferometric combinations of Sentinel-1 images for the small baseline subset (SBAS) analysis are shown in Figure 2. Precise orbit ephemerides (POD) data provided by European Space Agency (ESA) were used for the interferometric processing of the Sentinel-1 datasets. The digital elevation model (DEM) of 1 arcsec resolution (about 30 m) produced by the Shuttle Radar Topography Mission (SRTM) [32] is taken as a reference to facilitate differential interferogram generation and geocoding.

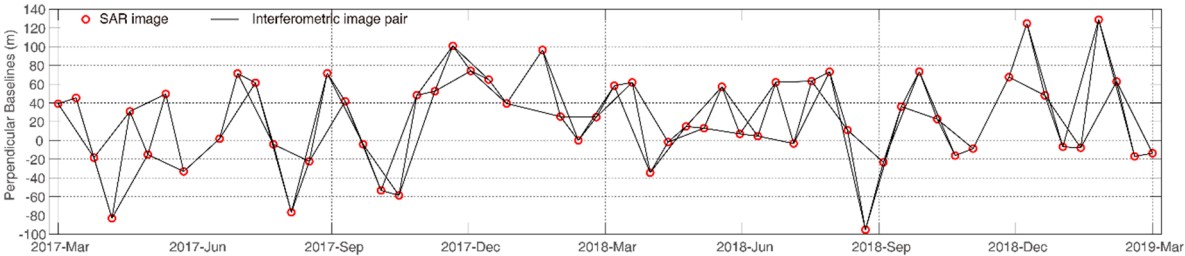

**Figure 2.** Interferometric combinations of Sentinel-1 datasets from path 11 for the time series analysis.

## 3. Method

### 3.1. Time Series Sentinel-1 InSAR Analysis

Assuming we have *N*+1 Sentinel-1 SAR images acquired in Terrain Observation with Progressive Scans (TOPS) mode, an image that maximizes the total correlation was first selected as a common master image according to the distribution of perpendicular baselines and temporal baselines. In this study, the image acquired at 13 February 2018 was chosen as the common master. Due to the nonstationarity of the squint angle during acquisition, there is a linear variation of the doppler centroid frequency in the SAR data. Thus, high level co-registration accuracy is required for Sentinel-1 InSAR analysis [33,34]. In our study, precise orbit data and external DEM are combined for geometric co-registration with respect to the master image. Enhanced spectral diversity (ESD) was then utilized to refine the co-registration results. Deramping and reramping are then carried out to prepare the images for the interferogram generation. In this study, images with temporal baselines shorter than 90 days are combined.

The amplitude dispersion value is used to initially select the point-like targets. A loose threshold, e.g., 0.6, can be set to include both persistent scatterers and slowly decorrelated pixels. The final pixels for time series analysis are determined by phase stability analysis [19,35]. Then, three-dimensional phase unwrapping was carried out to all the selected pixels [35]. The phase components in each pixel can be expressed as:

$$\varphi_{Unw} = \varphi_{orb} + \varphi_{topo} + \varphi_{disp} + \varphi_{aps} + \varphi_n \tag{1}$$

where $\varphi_{Unw}$ is the unwrapped phase, $\varphi_{orb}$, $\varphi_{topo}$, $\varphi_{disp}$, $\varphi_{aps}$, and $\varphi_n$ are phase components of orbital phase ramps, inaccurate topography, ground displacement and atmospheric disturbance and noise, separately. The displacement components can be further decomposed as a linear component and nonlinear component.

$$\varphi_{disp} = \Delta d \cdot \Delta t + \varphi_{nl} \tag{2}$$

*Δd* and *Δt* are the linear displacement rate and temporal baseline of the interferogram. $\varphi_{nl}$ stands for the nonlinear displacement.

Each component in the unwrapped phase can be estimated by the corresponding characteristics. The orbital ramp can be estimated with a bilinear model [36]. Inaccuracy of the reference DEM can be linearly estimated by the least square method with perpendicular baselines and unwrapped phases. Although the atmospheric disturbance is composed of ionosphere and tropospheric effect, we here only consider the tropospheric part with the reasonable assumption that the tropospheric effect on C-band InSAR observations over mid latitude regions is usually subtle and thus can be ignored. In our study, the tropospheric part is estimated with spatial and temporal filters. After removal of the above unfavorable phase components, the linear displacement can be estimated by least square fitting between phases and temporal baselines. The nonlinear part can then be obtained by singular value decomposition (SVD).

### 3.2. Rainfall Induced Slope Displacement Modeling

Rainfall is an important factor affecting the loess slope stability [2,12]. Seasonal rainwater recharge will cause the water storage variation [37] and change the pore pressure, shear strength. As a result, seasonal accelerations will occur on slopes [38]. In general, the displacement signals are correlated with rainfall in the temporal dimension [39]. Here, we attempt to employ a climate-driven displacement model [37] to reconstruct the relationship between rainfall and time series loess slope displacement. We assume the nonlinear displacement is directly proportional to the former displacement and the residence time of rainfall. The impacts from other factors (e.g., anthropogenic activities or tectonic movements) are directly neglected. According to the model, the evolution of landslide nonlinear displacement can be approximately expressed as:

$$d(t) = (d(t-1)) \cdot e^{-\frac{1}{\tau(t)}} + \beta \cdot P(t) \tag{3}$$

where $t$, $d(t)$, and $P(t)$ are the time vector, rainfall induced displacements, and precipitation vector at daily scale. $\beta$ is the calibrated scaling factor. Meanwhile, the nonlinear displacement vector can also be called the daily displacement rate vector. The values of rainwater residence time $\tau(t)$ mean the rainwater leaves the region quickly (small value) or slowly (large value) through runoff or evapotranspiration [37]. It can be expressed as a function of daily air temperature:

$$\tau(t) = a + b \cdot T_Z(t) \tag{4}$$

where $a$ and $b$ are positive coefficients to be solved for our model. $T_Z(t)$ is a transformation of the original daily air temperature $T(t)$ to make $\tau$ only sensitive to temperature changes when temperature is above 0 °C.

$$T(0) = \begin{cases} 0, & T < 0 \\ T, & T \geq 0 \end{cases} \tag{5}$$

A sigmoid transform as shown in Equation (6) is utilized to standardize the temperature in order to moderate the influence of extreme temperature. After the transformation, $T_Z$ varies between 0 and 1 along with the increase or decrease of temperature.

$$T_Z = 1 - \tanh\left(\frac{\text{mean}(T_0) - T_0}{\text{SD}(T_0)}\right) \tag{6}$$

Meanwhile, the initial value of displacement $d(0)$ can be determined by the analytical solution for the equilibrium state of Equation (3) given a mean precipitation $P$:

$$d(0) = \frac{\text{mean}(P)}{1 - \text{mean}(e^{-\frac{1}{\tau(t)}})} \tag{7}$$

The parameters are inverted with the obtained time series nonlinear displacements $d_{nl}$ in the last step together with the meteorological data including precipitation and air temperature at daily scale. The formulation is as follows:

$$d_{nl}(t) = d(t) + \varepsilon \tag{8}$$

where $\varepsilon$ is the error. The model parameters $a$, $b$ in Equation (4) and $\beta$ in Equation (3) are determined by using a Markov Chain Monte Carlo procedure [40]. The workflow of our method is shown in Figure 3.

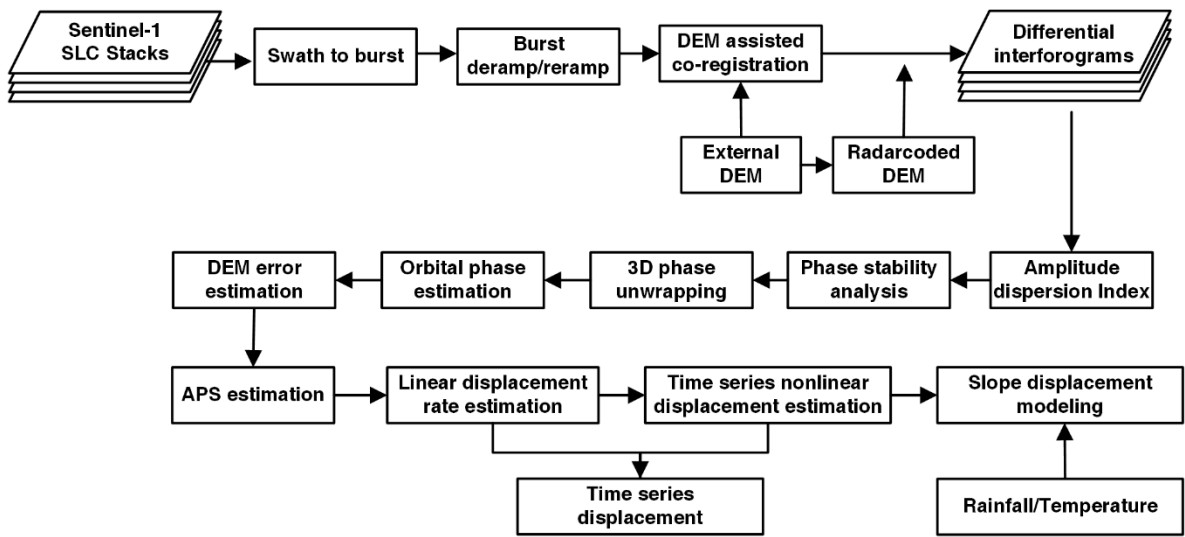

**Figure 3.** Workflow of time-series Interferometry Synthetic Aperture Radar (InSAR) analysis and displacement modeling in this study.

## 4. Results

### 4.1. Mining Area Detection from Differential Interferograms

Time series InSAR analysis only keeps those pixels that show a high level of coherence in all the interferograms. Coherent pixels in highly coherent interferograms might not be selected because of the low coherence of the same points in other interferograms [41]. As a result, detailed information in high coherence interferograms might be neglected. Therefore, analyses of differential interferograms are essential. Meanwhile, as we mentioned before, our study area is located in Shanxi Province with rich mineral resources. The mining of underground mineral resources inevitably creates the conditions for subsidence. The magnitude of subsidence generally decreases from the center to the edge of the exploited areas, forming subsidence funnels [42]. Two highly coherent interferograms shown in Figure 4 were selected as examples. The temporal baselines for both interferograms are 12 days, and the perpendicular baselines are 7 and –1 m, respectively. The normal subsidence funnels are of round or oval shape, which is very obvious in the interferograms shown in Figure 4a,c.

In our study, a total of 44 circular or elliptical deformation signals were detected. The positions of identified phase distortions marked with black solid polygons in each interferogram are a little bit different which might be induced by the evolution of continuous mining activities. Meanwhile, the magnitude of displacement, as shown in Figure 4b,d, varies in different periods because of the difference of mining rate or other factors. The detected number of mining areas is far less than expected. This might be the result of a series of decapacity policies to optimize the coal capacity since 2016 by reducing excessive capacity [43]. The quantity of coal mines and coal production have deceased since then. Shanxi Province is the largest coal decapacity province in the last three years [42]. Consequently, the mining activities might also slow down, resulting in subtle displacements on the ground, which might not be detectable on the short-period interferograms. Meanwhile, we only labeled fringes with typical circular or elliptical shapes as mining areas to avoid misinterpretation.

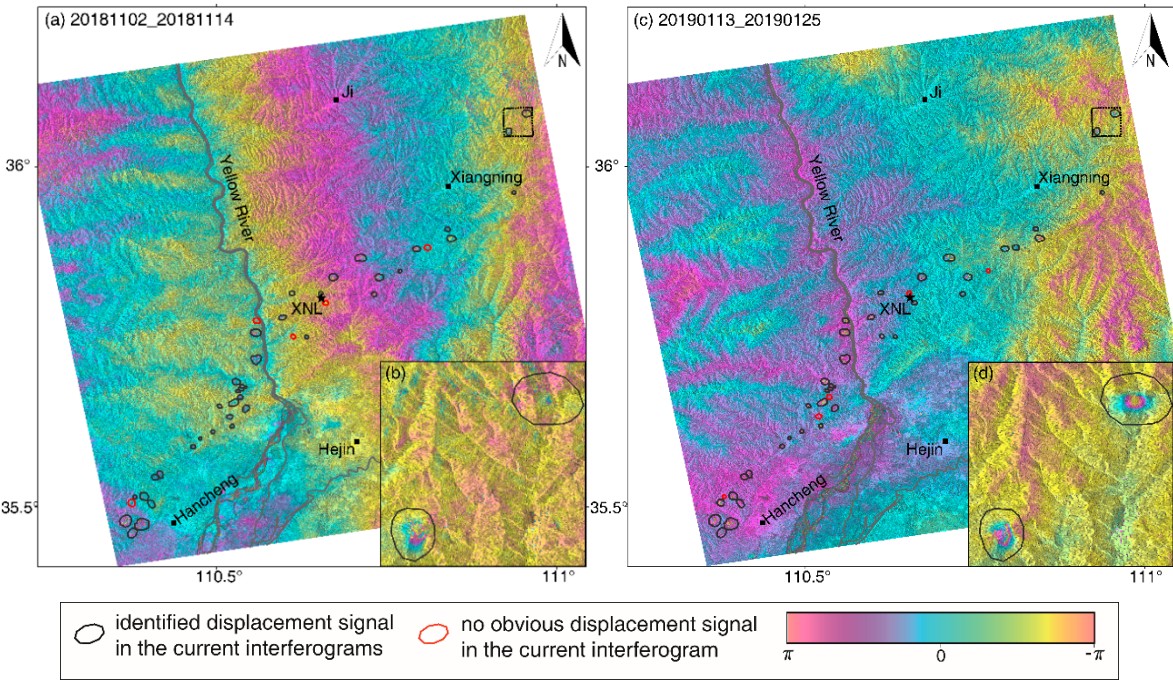

**Figure 4.** Differential interferograms produced from Sentinel-1 data pairs. (**a**) 20181102-20181114, (**c**) 20190113-20190125. (**b**) and (**d**) are enlarged interferograms marked with dashed rectangle in (**a**) and (**c**).

## 4.2. Mean Velocity Map Derived by Time Series InSAR Analysis

The mean displacement velocity map derived by time series Sentinel-1 InSAR analysis is shown in Figure 5. It is worth noting that the red color (negative value) indicates movement away from the satellite, while the blue color (positive value) indicates movement toward the satellite. A total number of 4227.531 measurement points (MPs) within the study area of 5600 km² were identified and analyzed. Although the color range in Figure 4a is set as from –40 to 40 mm/yr to better illustrate our results, there are a few MPs showing active deformations beyond this range. The most serious displacement reached over 144 mm/yr. Approximately 90% of the MPs show displacement rates between –10 and 10 mm/yr in Figure 5a, which suggests overall stability of our study area.

Figure 5b,c show the linear and nonlinear components of the line of sight (LOS) displacement rate, respectively. We can see clearly the subtle difference between the total displacement rate and its linear component. However, the maximum nonlinear displacement rates of ~15 mm/yr were detected in some unstable areas. This might be mainly correlated with seasonal rainfalls, which will be further analyzed in the following sections.

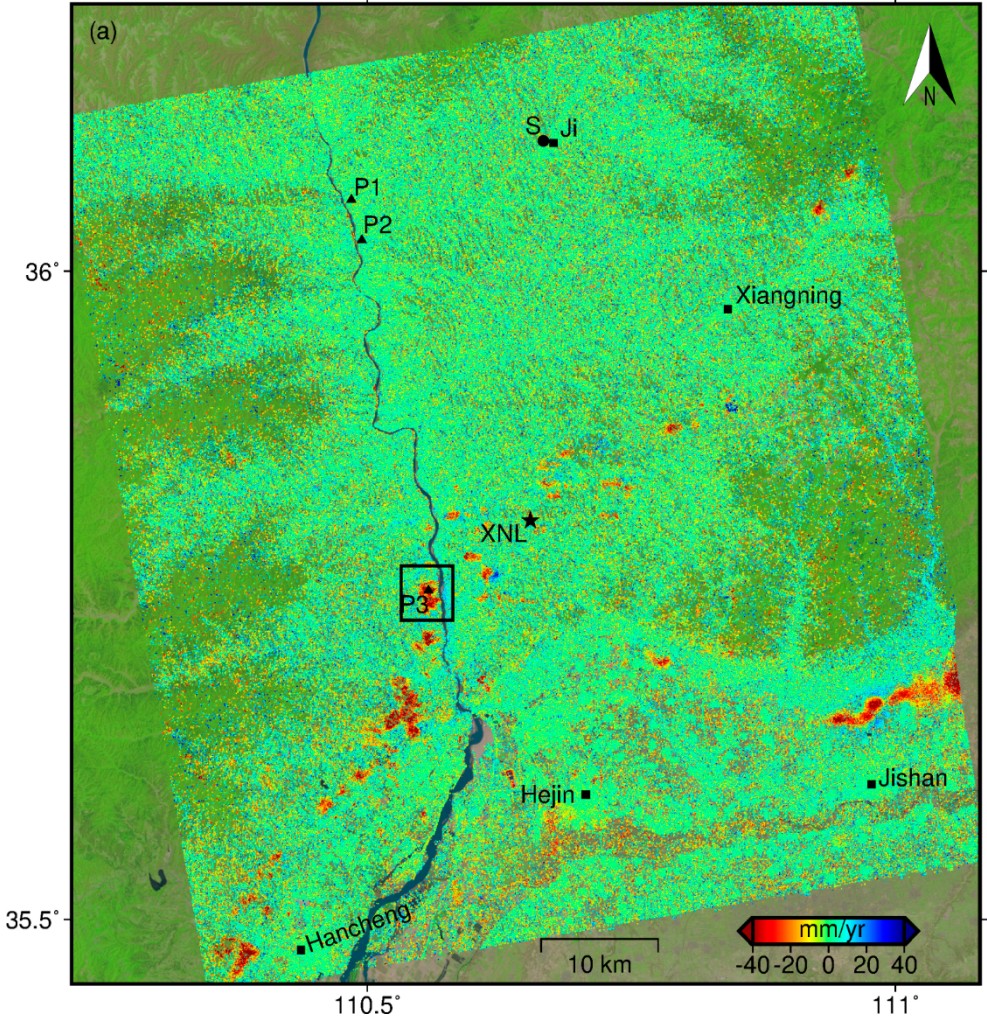

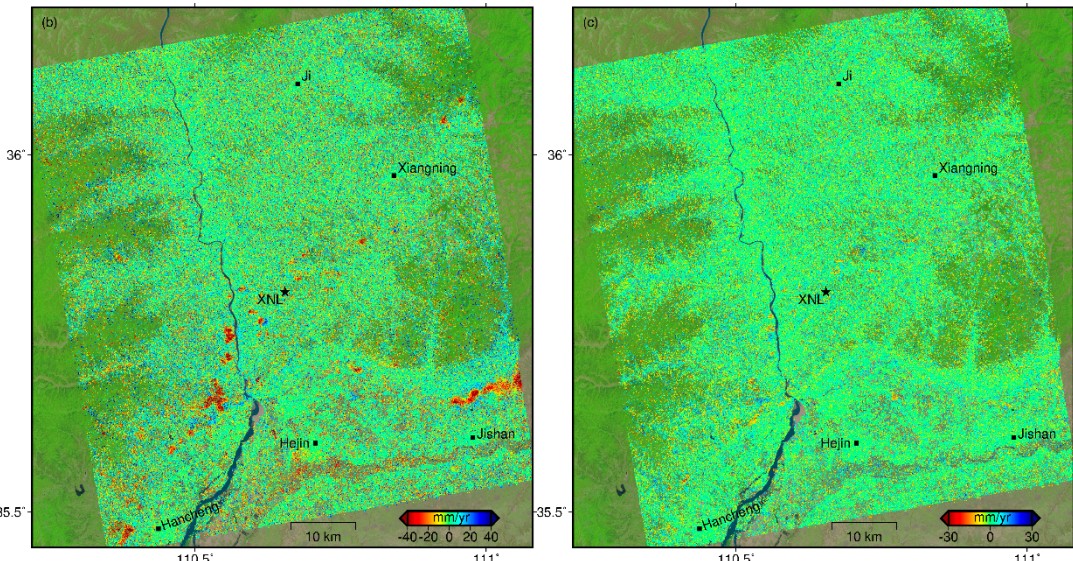

**Figure 5.** Displacement velocity map derived from Sentinel-1 datasets. (**a**) The total mean line of sight (LOS) displacement velocity, S marks the location of the weather station; (**b**) the linear component and (**c**) the nonlinear component. The background is a Sentinel-2 image covering our study area acquired on 5 June 2018. The black square outlines the mining induced displacement area to be analyzed in Figure 7. P1, P2, and P3 are selected points for the time series analysis in Figure 9.

### 4.3. Identification of Unstable Areas

A total number of 597 unstable areas covering 41.7 km² were detected from the mean displacement rate map, as shown in Figure 6. Those locations overlapped with the displacement signals identified by InSAR (shown in Figure 4) are listed as mining induced displacement signals. The large areas located at north Jishan were caused by extraction of ground water [28]. The most serious displacement rate reached approximately 90 mm/yr, which agreed in magnitude with the results in a previous study from Advanced Land Observing Satellite (ALOS) Phased Array type L-band Synthetic Aperture Radar (PALSAR) and Envisat ASAR datasets during 2007–2010 [28]. About 70% of the detected spots in deformation are located in landslide-prone mountainous areas. We roughly categorized the detected displacement signals into three classes, as shown in Figure 6. Those subsidence areas induced by mining and other anthropogenic activities within the flat Fenwei Basin are marked by blue polygons. Those unstable slopes associated with mining activities are rendered in red. Movements of other active slopes outlined by black polygons are mainly correlated with natural factors such as rainfall or changes of river water level.

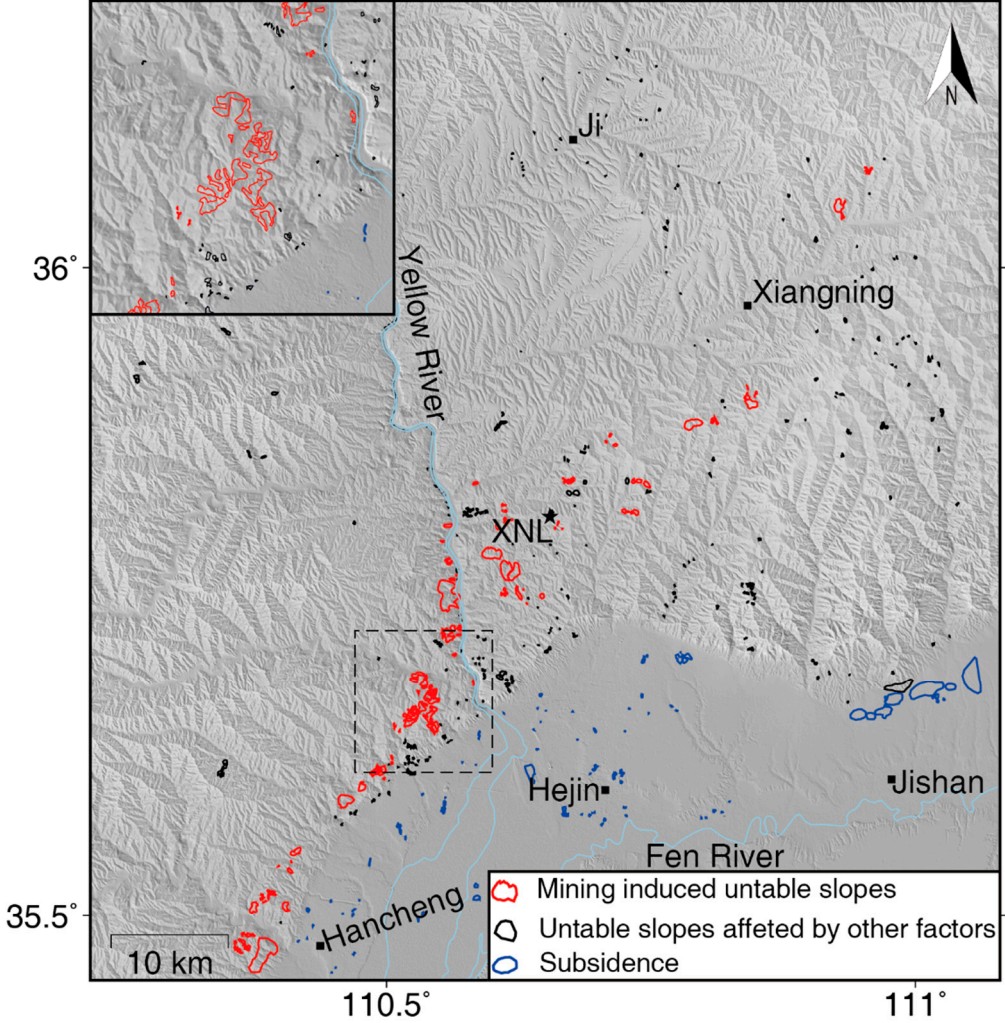

**Figure 6.** Detected unstable areas within the study area. The background is the hill-shaded Digital Elevation Model (DEM).

Multitemporal cumulative displacements of a mining induced unstable area outlined by the black square in Figure 5a are illustrated in Figure 7. The cumulated deformation increases in time as expected. The maximum accumulated LOS displacement reached more than 100 mm in about two years.

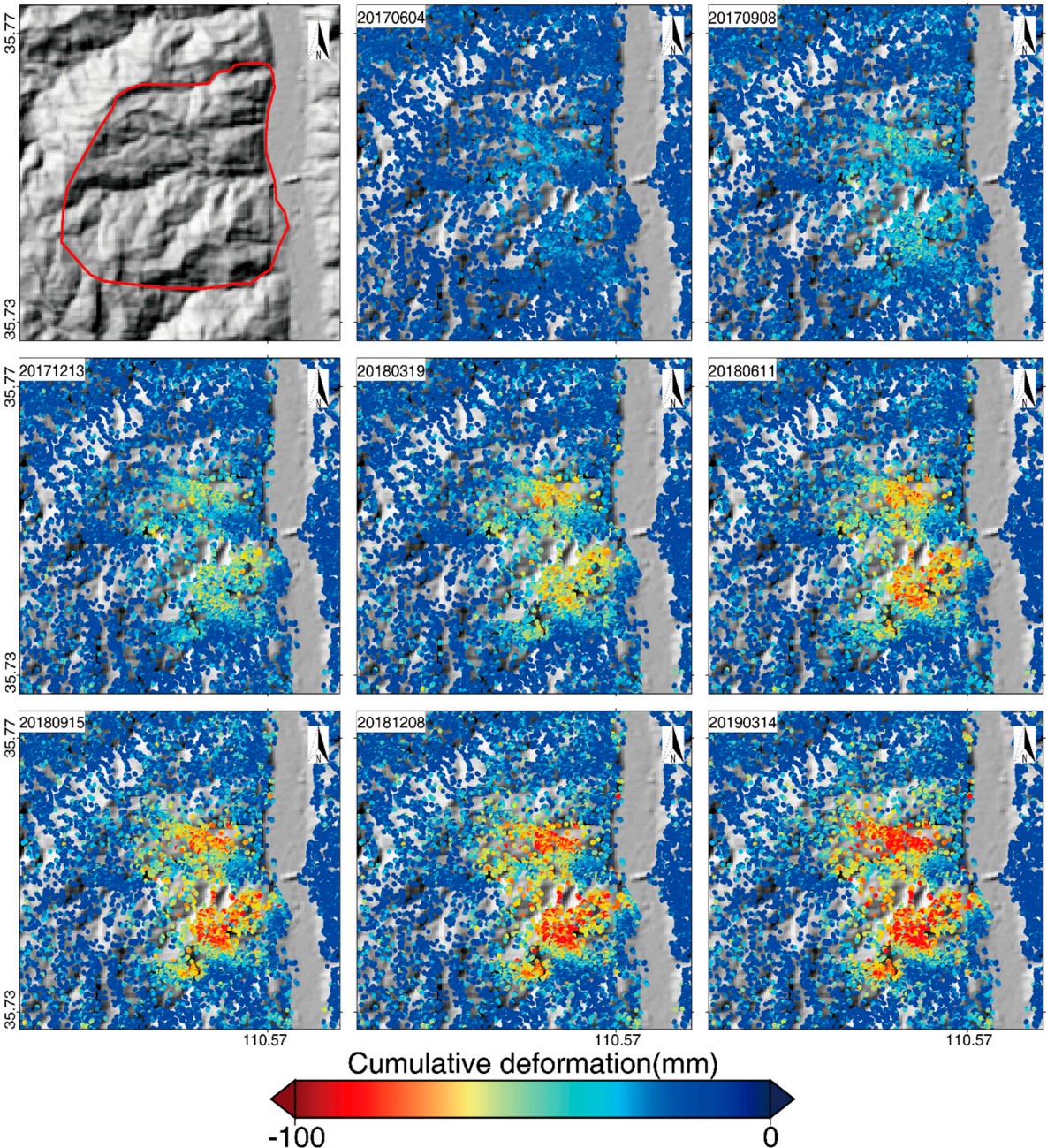

**Figure 7.** Accumulative LOS displacements of a slope selected from Figure 5 with respect to 20170302. Observation dates are annotated at upper-left corners of all subfigures.

## 5. Displacement Time Series Analysis and Discussion

### 5.1. Possible Cause of the Recent Xiangning Landslide Disaster

Figure 8a shows the mean LOS displacement velocity map of the Xiangning landslide from March 2017 to 2019 overlaid on an optical image acquired in February 2019. Considering the small spatial coverage, an averaged time series displacement calculated from all the pixels in the collapsed area was used to infer the possible cause of the recent failure event. The area outlined by the dashed line collapsed into the valley. As we can see from the mean velocity map, the displacement of detected pixels were almost stable. The results of differential InSAR and time series InSAR analyses firmly agree with the official story that the Xiangning landslide is not located at mining areas [44]. However, the true cause of the catastrophe has not been resolved and publicly announced until now.

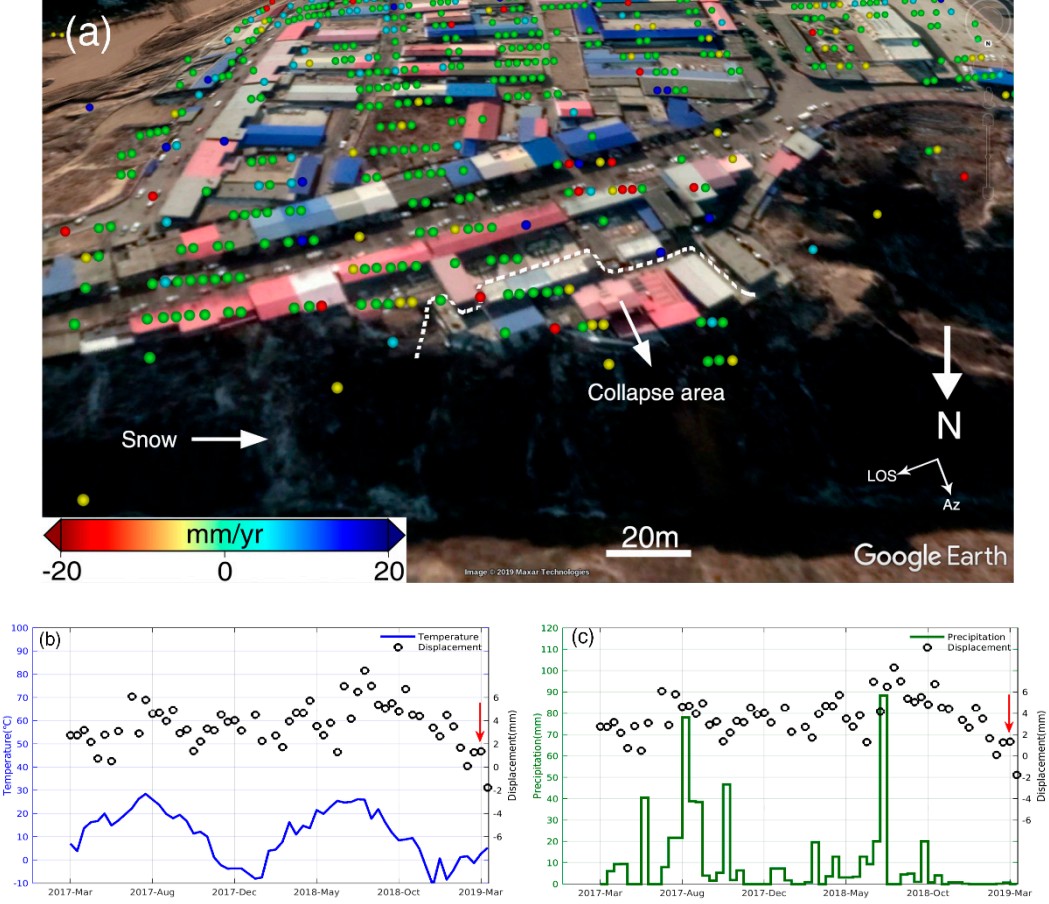

**Figure 8.** (**a**) Mean LOS displacement rates of the Xiangning landslide measured from Sentinel-1 datasets overlaid on Google Earth image obtained at 20 February 2019, the white dashed line represents the boundary between the collapsed area and stable area. (**b**) and (**c**) are the averaged time series displacement of the Xiangning landslide versus temperature and precipitation, respectively.

The deep valley areas are not favorable for InSAR observation, and unexceptionally no MP was identified in the valley, as shown in Figure 8a. Meanwhile, sunlight can hardly reach this area because it is a north oriented steep slope. Therefore, the winter snow could be frozen for a long time. Although there was almost no precipitation since December 2018, signs of cumulated snows were identified in the valley as inferred from the optical image acquired on 20 February 2019 in Figure 8a. However, the temperature in our study area rose to above 0° at the beginning of February 2019.

Generally, there are lag effects between temperature changes and freezing/thawing activity [45]. The status of the Xiangning landslide remained stable until 2 March 2019. A sudden acceleration was identified, which might be caused by the thawing activities in the loess, as shown in Figure 8b,c. Previous studies showed that the freeze and thawing activities are extremely harmful to the stability of loess slopes [4,46]. In general, the strength of frozen loess is much higher than that of natural loess. However, the strength will be significantly reduced when the snow or ice melted, possibly leading to the failure of the Xiangning landslide. The failure of the slope might start from the edge area, according to the overall stability of MPs on the landslide before collapse. The melted ice or snow could decrease the cohesion and frictional force inside the slope, and then break the limit equilibrium to cause the failure. However, more evidence should be collected in our future study to validate this inference.

### 5.2. Slope Displacements Correlated with Rainfall

Concentrated seasonal rainfall is a key impact factor that may trigger or accelerate the movements of loess landslides [2,3]. Three points P1, P2, and P3 marked in Figure 5a are selected to explore the relationship between displacements and precipitation. Mean LOS displacement velocity

of these points are given in Figure 9a–c. The mining induced displacement at P3, as shown in Figure 9c, obviously covers a larger area. Figure 9d–f show the time series displacements of P1, P2, and P3 while Figure 9g–i only plot the nonlinear components of displacements. After removal of the linear trend, significant periodic evolution patterns of displacements can be observed at these points. The periodic displacements are almost synchronized with the seasonal rainfall.

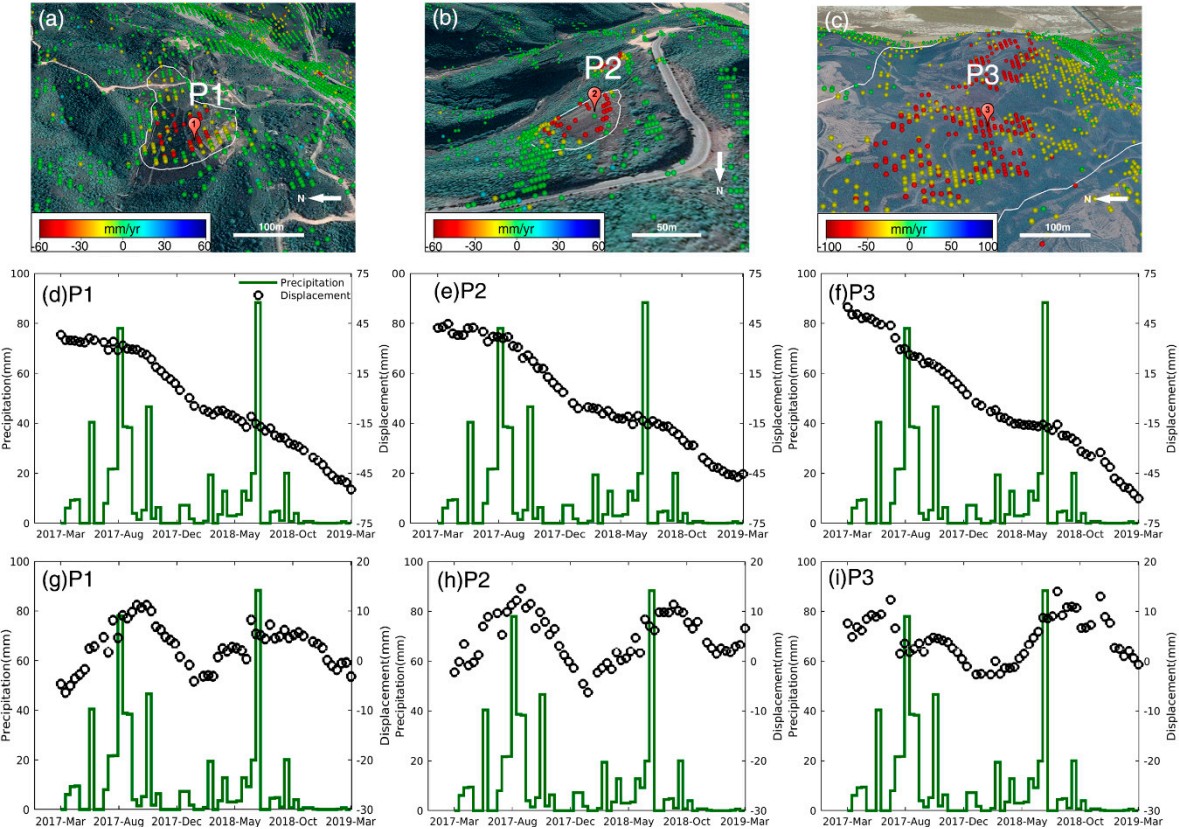

**Figure 9.** (**a**)–(**c**) mean LOS displacement velocity maps overlaid on Google Earth™ maps. (**d**)–(**f**) time series displacements of P1, P2, and P3 versus rainfall and (**g**)–(**i**) nonlinear displacements of points P1, P2, and P3 versus rainfall.

To further explore the evolution mechanism of such displacements, we applied the climate-driven displacement model described in Section 3.2 to try building the quantitative linkage between nonlinear displacements and meteorological data (precipitation and temperature) recorded by a weather station nearby (longitude 110° 40' east and latitude 36° 06' north). The modeling results are visualized in Figure 10. Good agreements were achieved for P1 and P2 with correlation coefficient $R^2$ higher than 0.7, as shown in Figure 10d,e. However, the correlation coefficient $R^2$ for P3 is only 0.15, indicating the low reliability of our model. The precipitation and temperature data have uncertainties on these three points located in different regions. We found that the distances between that weather station and P1, P2, and P3 are approximately 17.1, 17.7, and 39.7 km, respectively, which suggests there might be big differences between the meteorological measurements and the real values of precipitation and temperature at P3. There are also limitations of this model, as we only consider the precipitation and temperature. We do not consider other impact, e.g., net radiation and humidity. In addition, it is reasonable to hypothesize that the temporal variations of coal mining intensity should have a significant impact on the nonlinear displacements at P3 located within a coal mining area.

The resolved optimal values of model parameters *a*, *b*, and *β* are 36.84, 155.18, 0.03 for P1 and 11.20, 63.43, 0.04 for P2. The residence time of rainwater varies a lot as shown in Figure 10a,b, which might depend on the slope materials and structures, etc. The shear strength of the loess will be reduced when infiltration occurs. This will further induce cracks or cause hidden cracks to reopen at

the surface [7]. Meanwhile, rainwater can infiltrate where sinkholes, cracks, or weathering fractures are presented. The shear strength of the loess slopes will then be reduced with the rise of groundwater level. There will be lags between displacement and precipitation during infiltration and shear strength variation process. In addition, precipitation will increase loess weight.

The effects of rainfall on displacements at P1, P2, and P3 show similar patterns. The differences in periodic fluctuation amplitude of displacements might be related to the material composition of the slope body, the depth of rainfall infiltration, and the actual rainfall on the slope.

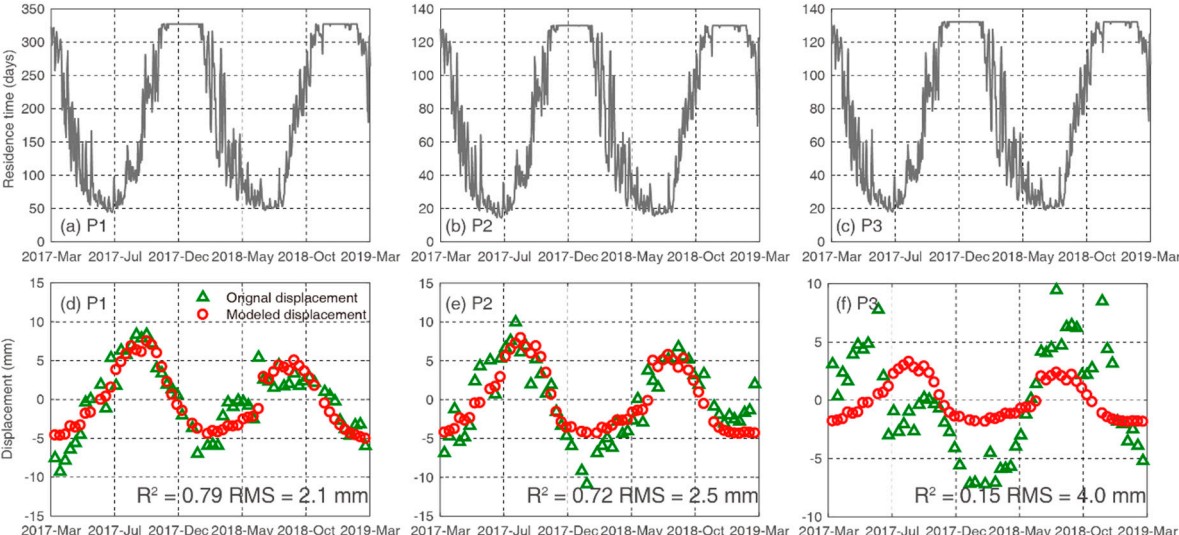

**Figure 10.** (**a**)–(**c**) The residence times of rainwater at P1, P2, and P3; (**d**)–(**f**) InSAR-measured versus modeled nonlinear displacements at points P1, P2, and P3.

Moreover, our study area is a traditional agricultural region in Shanxi Province. Intensive irrigation in the loess areas poses great threats to the stability of loess slopes [10]. The dramatic increase of groundwater level induced by irrigation will seriously decrease the shear strength of loess. Meanwhile, the mining process may not only create voids beneath the ground, but also make the hydrological conditions more complex [28].

## 6. Conclusions

Loess slopes are very fragile subject to the intensified anthropogenic activities and natural impact factors. In this study, we investigated the stability of loess slopes in Xiangning County, Shanxi Province, and its surrounding areas with the time series Sentinel-1 InSAR analyses. A total of 597 unstable spots covering 41.7 km$^2$ were identified. However, the side-looking geometry limited the visibility of SAR datasets in mountainous areas. Combining the use of datasets from ascending and descending orbits will increase the visibility to some extent. Moreover, it is necessary to comprehensively use the results from InSAR, optical images interpretation and field investigation to maximize the unstable slopes detection coverage.

In general, there might be precursory signals before slope failure events [17,47]. However, displacement rates of most MPs identified on the Xiangning landslide before collapse are less than 10 mm/yr. This brought great challenges to the identification of such landslides and disaster early warning. Mean displacements of MPs in the collapsed areas might provide clues for the cause of the catastrophe. More information should be collected to validate our assumption.

By separating the nonlinear and linear components of LOS displacement measurements from each other, displacement accelerations that coincide with seasonal rainfalls can be clearly identified on loess slopes. A climate-driven displacement model was employed to try building the linkage between nonlinear displacements and meteorological factors (precipitation and temperature here). In general, the modeled nonlinear displacements agree well with InSAR measurements in tendency. Rainfall induced water content changes in loess will significantly affect the stability of loess slopes.

However, this model needs precipitation and temperature measurements close to the slope. In our study, only sparse in-suit observation was obtained to study the relationship between displacement and climate-driven factors. It is hard for us to apply this model to all the active slopes. In the future, high precision climate factors measurements from satellite might be used as alternatives. In addition, this model can be used for short term landslide displacement forecast.

With the continuous accumulation of Sentinel-1 data, it is convenient for us to pay close attention to the disaster-prone loess areas. The routine updating of mean displacement velocity maps is an urgent but difficult task that brings big challenges to the InSAR community. However, it is not enough to identify landslides with limited precursor information such as the Xiangning landslide. Combination of multi-source results such as displacement maps, landslides susceptibility maps, and geological maps are necessary.

**Author Contributions:** X.S., L.Z. (Li Zhang), Y.Z., L.Z. (Lu Zhang), and M.L. conceived and designed the experiments; X.S. and L.Z. (Li Zhang) performed the experiments; Y.Z. analyzed the results; X.S., L.Z. (Li Zhang), and Y.Z. wrote original manuscript; L.Z. (Lu Zhang) edited the manuscript. All authors have read and approved the final manuscript.

**Funding:** This work was financially supported by the National Key R&D Program of China (Grant No. 2017YFB0502700), the National Natural Science Foundation of China (Grant No. 41702376, 41774006), the Provincial Key R&D Program of Sichuan Ministry of Science and Technology (Grant No. 2019YFS0074), the CRSRI Open Research Program (Grant No. CKWV2018482/KY), and the Provincial Key Technology Research and Development Program of Sichuan Ministry of Natural Resources for Ecological Geohazard Prevention and Mitigation in the "8•8" Jiuzhaigou Earthquake Area (Grant No. KJ-2018-21).

**Acknowledgments:** The Sentinel-1 datasets were provided by the European Space Agency (ESA) freely through the Sentinels Scientific Data Hub under the ESA-MOST Dragon 4 Program (id 32278). The precipitation and temperature data are provided by the China Meteorological Data Service Center (http://data.cma.cn).

**Conflicts of Interest:** The authors declare no conflict of interest.

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
