# Peer review of "Detection and Characterization of Active Slope Deformations with Sentinel-1 InSAR Analyses in the Southwest Area of Shanxi, China"

_remotesensing, doi:10.3390/rs12030392_

Round 1

Reviewer 1 Report

Summary

The manuscript is devoted to important problem of identifying the potential active loess landslides and mapping active loess slopes with Sentinel-1 InSAR analyses.

Broad comments

The relevance of the problem is not in doubt and the obtained results could help to reduce the damage from possible future displacements. The study is performed and presented at a high scientific level.

Specific comments

Despite the overall high level of the study, there are the following comments.

In Introduction section much attention is paid to describing the relevance of the study. However, modern approaches to solving the problem of loess slopes mapping using SAR data are not sufficiently discussed. Please provide a brief overview of the achievements in the field of slope stability monitoring using InSAR, as well as to determine the place the Authors’ research in the context of this problem, indicate the uniqueness of research.

There is no doubt the practical novelty of the manuscript, but the scientific novelty is not clearly formulated. The Authors should focus on the methods and approaches improved in this work, on the uniqueness of this work in the context of up-to-date remote sensing research.

What is the accuracy of surface movements estimation? Please describe in more detail the ground control samples on which the results were verified? How would using the data from another satellite affect the result?

Line 10 “Correspondence”: duplicate word.

Line 90 Figure 1: the title of the Figure 1 is very detailed and difficult to understand. It would be more convenient for a reader to represent the title in the form of a legend.

Line 102 Figure 2: curves are merged in the Figure 2. What conclusions should a reader draw looking at Figure 2?

Line 120: “where”: please use the lowercase and no paragraph spacing.

Line 226 “caused by mining activities”: were the results of displacements assessment for these areas verified using ground-based observations?

Figure 1, 3, 4, 5: north pointer is missed.

The manuscript should be presented in such a way that the reader could reproduce the calculations and compare his results with the results presented in the manuscript. For this please supplement the third section with a description of the displacement maps constructing process and flowchart. For example, please see Figure 2 of this paper:
https://www.mdpi.com/2071-1050/11/18/5090

Author Response

Authors used InSAR data to study the ground deformation in Shanxi (China). They also compared the deformation with the seasonal changes of temperatures and rain falls. They found a good agreement with their deformation model driven by climate variations with the non linear component of the displacement. The paper have some weakness.

You use only one looking geometry and so you are not able to discriminate between vertical or horizontal displacements. However you classify the observed signals in natural or anthropic slope instability even if some of the examples that you show seems to be located in flat areas. The majority of the observed signals could be related to simple subsidence that can produce a sinkhole but not the slope instability. I suggest to plot the instability areas on a slope map that will allow to better characterize the type. Another thing on the classification you used, circular and elliptical deformations could be also natural and at the same time you can have anthropic signal with other shapes. Another explanation could be very localized artifacts due to layover effect. In the area you have rapid topographic changes that could produce this small signals.

Ans.: Indeed, we are facing the problem of data availability. Combining use of measurements from ascending and descending datasets will definitely help us to discriminate between vertical or horizontal displacements. Now, only Sentinel-1 datasets in ascending orbits are acquired in most areas of China.

Actually, we produced the slope instability map according to the topography model. The figure 9(b) might be a little confusing. Thus, we reproduced this Figure 9(b).

It is true that the circular and elliptical deformations could be also natural and at the same time you can have anthropic signal with other shapes in the final displacement rate map. However, here we only identify the obvious circular and elliptical fringes on the differential interforograms instead of from the final displacement rate maps. The final results from differential InSAR and final displacement rate maps are combined. Localized artifacts due to layover will get total decorrelated, thus will be excluded in the final results.

Another important concern I have is on the amplitude of the deformation that you show in the deformation map. You showed two interferogams with some areas that displace of >2cm in 12 days. Since the displacement is mainly linear I would expect a displacement rate up to 60 cm/yr while you obtain 4 cm/yr.- Detection (or identification) and characterization of active slope deformations with Sentinel-1 InSAR analyses in southwest area of Shanxi, China.

Ans.: We compared our results from Sentinel-1 and from Envisat ASAR and ALOS PALSAR from Zhao et al. (2018) from 2007 to 2010. Although the temporal coverage is different, the retrieved displacement rates are very close. The maximum displacement rate reached over 114 mm/yr. Besides, the displacements induced by mining activities are correlated with the evacuation rate. The fringes will gradually get blurred along with the evolution of evacuation. Unavoidable phase unwrapping errors will occur during the time series InSAR analysis process. Thus, there might be some under estimation in some of the mining areas. This is a problem that we are working on now.

The title of our paper has been revised.

Zhao, C.; Liu, C.; Zhang, Q.; Lu, Z.; Yang, C., Deformation of Linfen-Yuncheng Basin (China) and its mechanisms revealed by Π-RATE InSAR technique. Remote Sensing of Environment 2018, 218, 221-230.

- Line 14. Replace ’20 fatalities’ with ‘fatalities of 20 people’.

Ans.: Agreed and revised (Line 15).

- Line 14. Rephrase ‘sounded the alarm’.

Ans.: Agreed and revised (Lines 15-16).

- Line 17. It is a good tool for some things, but not "ideal". Please, avoid overselling.

Ans.: Agreed and revised. We change the “ideal” into “effective” (Line 18).

- Line 22. ‘activities’ substitute with ‘cycles’.

Ans.: Agreed and revised (Line 23).

- Line 30. Replace ‘is a special landform formed by aeolian transport, loess accumulation and erosion in the’ with ‘is an aeolian deposit that was accumulated over the’.

Ans.: Agreed and revised (Line 31).

- Line 34. Replace ‘when getting wetted with water’ with ‘with the increasing water content’.

Ans.: Agreed and revised (Lines 34-35).

- Line 36. It is not the distribution which is affected by the predisposing and triggering factors listed. Please rephrase ‘the distribution of these landslides are affected by’.

Ans.: Agreed and revised (Line 37).

- Line 42. ‘a lot of’. How many? Provide at least an order of magnitude. Please avoid such generic expressions, such as ‘a lot of’.

Ans.: The loess landslide in the terraces areas are found retrogressive. Here we use “retrogressive” instead of specific numbers (Line 43).

- Line 44. Delete ‘unfortunately’.

Ans.: Agreed and revised.

- Line 46. ‘Displacements of slopes are direct indictors of landslide stability’. This sentence tautologic.

Ans.: We removed this sentence.

- Line51. be more specific on the type of landslides (which is directly linked to their range of velocity), and add references. Faster landslides are detectable using the amplitude images. I would suggest adding this to the references to widen the spectrum of the landslides that can be detected using SAR: Mondini, A., Santangelo, M., Rocchetti, M., Rossetto, F., Manconi, A., Monserrat, O., 2019

Ans.: Agreed and revised (Lines 51-54).

- Line 60. ‘The second part describes the SAR image processing algorithm followed by the introduction of study area and datasets. The analyses of the results and discussions are presented in section 4 and 5. The conclusions are drawn in the end.’ . Do not start with the second session. Just " chapter 2 deals with"...and so on

Ans.: Agreed and revised (Lines 74-76).

- Line 67. Replace ‘special geomorphological’ with ‘peculiar Lithological’.

Ans.: Agreed and revised (Line 80).

- Line 68. Replace ‘vulnerable’ with ‘prone or susceptible’. Vulnerability has to do with exposed elements and expected damage. See comments on terminology.

Ans.: Agreed and revised (Line 81).

- Line 78. Replace ‘with an altitude of 300-2000 m. The 78 landforms are characterized as loess terrace, loess ridge and loess hill’ with ‘hillslopes with an altitude ranging between 300 and 2000 m a.s.l.’

Ans.: Agreed and revised (Lines 91-92).

- Line 80. Replace ‘loose’ with ‘unconsolidated’.

Ans.: Agreed and revised (Line 93).

- Line 80. Replace ‘The erosions are’ with ‘Erosion is’.

Ans.: Agreed and revised (Line 93).

- Line 86. ‘The freezing period and the depth of frozen soil vary with the terrain and location.’ Be quantitative, please.

Ans.: The general annual freezing period is from October to March, however, we don’t have data of the depth of the frozen soil now (Line 103).

-Figure 1. XNL black star. Please fill it also with black, so it is more visible.

Ans.: Agreed and revised (Figure 1).

- Line 96. Replace ‘was given’ with ‘is shown’.

Ans.: Agreed and revised (Line 109).

- Figure 2. what are the red circles? Please add a legend.

Ans.: Agreed and revised (Figure 2).

- Line 114. Replace ‘less’ with ‘shorter’.

Ans.: Agreed and revised (Line 129).

- Line 138. Delete ‘impact’.

Ans.: Agreed and revised.

- Line 140. ‘seasonal displacement accelerations of slopes at the same time’. Not comprehensible. Please rephrase and explain better.

Ans.: Agreed and revised (Lines 155-156).

- Line 143. ‘assume’. What are the implications of this assumption? What are we going to neglect? Please make the reader aware.

Ans.: This means we only consider the interaction between precipitation and nonlinear slope displacement. The impact from other factors (e.g. anthropogenic activities or tectonic movements) is directly neglected (Lines 160-161).

- Line 166. Replace ‘following’ with ‘follows’.

Ans.: Agreed and revised (Line 183).

- Line 172. Replace ‘remain’ with ‘show’.

Ans.: Agreed and revised (Line 192).

- Line 178. Replace ‘voids, making the above ground sink’ with ‘the conditions for subsidence.’

Ans.: Agreed and revised (Line 198).

- Line 179. Replace ‘voids’ with ‘exploited areas’.

Ans.: Agreed and revised (Line 199).

- Line 188. Delete ‘impact’.

Ans.: Agreed and revised.

- Line 189. ‘de-capacity’ what does it mean? Please explain this word.

Ans.: In recent years, affected by factors such as slowing economic growth and energy structure adjustment, coal demand growth was significantly below expectation, supply capacity has remained excessive and supply and demand have fallen seriously out of balance, resulting in a general decline in corporate profits, chaotic market competition and increased safety production hazards. To reducing excessive capacity, the quantity of coal mines and coal production have deceased since 2016. We briefly explained this in the text (Lines 210-211).

- Figure 3. letters too small. Add legends with color scale and polygons colours.

Ans.: Agreed and revised (Figure 4).

- Line 214. Replace ‘at’ with ‘on’.

Ans.: Agreed and revised (Line 234).

- Figure 5. (1)Add legend and a zoomed map. (2) Center the zoom map here.

Ans.: Agreed and revised (Figure 6).

- Line 222. Title 4.3. Replace ‘Identification of unstable areas.’ with ‘Identification of unstable areas.’.[ Correct this throughout the text. In this paper the word mapping is not properly used (see general comments). Identification and detecion is more appropriate. ]

Ans.: Agreed and revised (Line 240).

- Figure 6. It would be nice to see an image without measures just to see a detail of the morphology.

Ans.: Agreed and revised (Figure 7).

- Line 240. Replace title 5’ Analyses of Time Series Displacements and Discussions’ with ‘Diplacement time series analysis and discussion’.

Ans.: Agreed and revised (Line 260).

- Figure 7(b)-(c). Add legend like in figure 8d-i.

Ans.: Agreed and revised (Figure 8).

- Line 258. ‘February 20, 2019.’ Refer to Figure 7a here.

Ans.: Agreed and revised (Line 279-280).

- Line 269. Replace ‘tragedy’ with ‘failure/collapse’.

Ans.: Agreed and revised (Line 290).

- Figure 8(a)-(c). letters, north arrows and scale should be far more visible.

Ans.: Agreed and revised (Figure 8a~c).

- Line 296. Replace ‘depends’ with ‘depend’ and ‘tensile’ with ‘shear’.

Ans.: Agreed and revised (Line 323).

- Line 297. Replace ‘rainfall penetrates into it’ with ‘infiltration occurs’.

Ans.: Agreed and revised (Line 324).

- Line 298.  Delete ‘downward’.

Ans.: Agreed and revised.

- Line 300. ‘Time lags will emerge during these processes.’ Not clear. Explain more.

Ans.: Agreed and revised (Line 327-328).

- Line 305. Replace ‘assume’ with ‘hypothesize’.

Ans.: Agreed and revised (Line 318).

- Line 315. Replace ‘In additional’ with ‘In addition’.

Ans.: We removed this sentence (Line 328).

- Line 316. Delete ‘fundamental’.

Ans.: We removed this sentence.

- Line 316. ‘spatial pattern of geodynamics across the whole Fenwei Basin’ . This is really not clear. Please rephrase taking care to clarify this concept. Otherwise delete the sentence.

Ans.: Agreed and revised. We removed this sentence.

- Line 328. ‘tracing the catastrophe’. Really unclear.

Ans.: Agreed and revised (Line 354).

Reviewer 2 Report

Presented paper is quite well composed,the topic is actual and described methods (especially rainfall iduced displacement modelling) include added value.  My comments are regarding mainly minor changes of pictures and some parts of text.

Figure 1 and surrounded text: It is not clear, what type of land cover dominates in the study area? The map does not show anything except topography and it is not mentioned in the text. The land cover is crucial for interfermetry.

 Line 71: "junction of Shanxi Province and Shaanxi Province" Is it right?

Chapter 2.2 and Figure 2: Why are the acqusitions not in Star Graph with master image in the middle? It would be much better. The date of the Master is not mentioned.

Chapter 3. 1. The description of the method is a liitle bit disarranged. I wonder, what kind of software was used?

Figures 3 and 5. A map legend would be better than description by words under the figure.

Figure 9. Why differ the timelines of parts a b c from d e f? 

Author Response

Presented paper is quite well composed, the topic is actual and described methods (especially rainfall iduced displacement modelling) include added value.  My comments are regarding mainly minor changes of pictures and some parts of text.

Figure 1 and surrounded text: It is not clear, what type of land cover dominates in the study area? The map does not show anything except topography and it is not mentioned in the text. The land cover is crucial for interfermetry.

Ans.: Agreed and revised. InSAR are sensitive to vegetation coverage. Thus, a NDVI map produced from a Sentinel-2 observation on 5 June 2018 is added as Figure 1(b).

Line 71: "junction of Shanxi Province and Shaanxi Province" Is it right?

Ans.: Agreed and revised. Our study area lies on the border between Shanxi and Shaanxi Province (Line 84).

Chapter 2.2 and Figure 2: Why are the acqusitions not in Star Graph with master image in the middle? It would be much better. The date of the Master is not mentioned.

Ans.: This is because the Small Baseline Subset (SBAS) InSAR method was employed in our study. Figure 2 shows the SAR interferometric combinations used in our study. All the images were co-registered to a common master, which was specified in the text. A star graph with a master image in the middle shown as below cannot elaborate the combinations. Therefore, we kept the original Figure 2 and elaborated the selected master image in Lines 122~123.

Chapter 3. 1. The description of the method is a liitle bit disarranged. I wonder, what kind of software was used?

Ans.: We rearranged section 3. Actually, we processed data with our own software. The method of Sentinel-1 interferometric data processing was published in the journal Computer & Geosciences, which is added as a reference.

Yu, Y.; Balz, T.; Luo, H.; Liao, M.; Zhang, L., GPU accelerated interferometric SAR processing for Sentinel-1 TOPS data. Computers & Geosciences 2019, 129, 12-25.

Figures 3 and 5. A map legend would be better than description by words under the figure.

Ans.: Agreed and revised (Figure 4 and 6).

Figure 9. Why differ the timelines of parts a b c from d e f?

Ans.: According to our assumption in section 3.2, the rainfall will residence in the slope for a while. In our study, the precipitation and temperature data records during a longer period from 2015 to 2019 than SAR observations are acquired as input to model the relationship between rainfall and displacements. In the revised manuscript, we have unified the timelines of all the subfigures to avoid misunderstanding (Figure 10).

Reviewer 3 Report

I have been asked to act as reviewer of the manuscript “Mapping active loess slopes in southwest area of Shanxi, China with Sentinel-1 InSAR analyses” by Xuguo Shi, Li Zhang, Yulong Zhong, Lu Zhang, Mingsheng Liao.

Most of my comments, even some general ones and suggestions, are contained as notes in the attached pdf. This letter only contains general comments.

The manuscript aims at presenting an approach to detect active deformations in slopes which lithology is exclusively consisting in loess, with particular focus on tryiing to identify precursors of failure in an event recorded in March 15th 2019. The authors also attempt to link simple climatological variables (precipitation, temperature) to the deformation pattern. In general, the idea proposed is interesting, despite not really novel. It is much a case study in a particular geological setting.

Despite the interesting idea, for how it is presented, the paper has several limitations and it can not yet be published as it is on the journal Remote Sensing unless moderate revisions are carried out to improve the publication and the work behind it.

The English language is, in general, good, even though some problems are present in a few expressions that the authors will find highlighted in the annotated pdf.

On the terminology, I noticed some confusion between some terms that sould be used more consistenly to the scientific Literature. The very first definition given of loess is “a special landform formed by aeolian translport, loess accumulation and erosion ...”. The definition has two problem. The first is logical: the explanation contains the explained word. So it comes out that loess is formed by loess. The second is geological: loess is not a landform but a lithology, and specifically an aeolian deposit. Furhtermore, dealing with terminology, I would delete from the manuscript all the subjective judgements. So words such as special, tragedy, etc. They are not objective and should be avoided in academic English. The individual points were highlighted in the pdf.

The title could be chosen better. Slopes are not active or whatever. They are just slopes. Processes can be active, such as deformations or landslides, for example. Furthermore, I find not really suitable the use of the term “mapping”, whereas “detection” would be much more appropriate in this case. Detection and mapping are not synonyms. The detection has to do with the identification that a landslide is there. Mapping has to do with a formal delineation of a landslide that has been detected. This difference is substantial, and the use of these words as synonims leads to some consistency issues and, above all, makes the reader a promise that is not kept throughout the paper. I would suggest the title: “Detection (or identification) and characterization of active slope deformations with Sentinel-1 InSAR analyses in southwest area of Shanxi, China”.

The structure of the paper is acceptable in general. Only two main comments. The authors should focus more on highlighting adantages and limitations of the technique, future development, how they place this work in the state of the art, what is the research question they are answering, and to what extent the question can be said answered. What other question are open through this work.

A more conceptual issue for me has to do with the type of failure the authors try to find precursors for. The loss lithology, as an aeolian deposit, has a high isotropy and the grain size is quite homogeneous since the transport action is carried out by wind, which sorts the grains based on the current intensity. Similarly to other deposits that share these characteristics, this type of weak, unconsolidated and uncohesive lithologies show little to no precursors before failures, which makes them poorly suitable to monitoring. A confirmation of this is what you can see in the plots of Figure 7, where there is evidence of a very little deformation before the failure that caused the death of people. I would say that this approach is not suitable for setting up an early warning system, contrarily to what the authors state in the abstract and introduction. The best approach to keep low vulnerability to population is preventing them from living on the edge of escarpments of loess plateau, where the river incision, and scarp retreat inevitably pose a high risk to the population. For me the analyses show that this technique is not feasible, at least in this band. The authors should state this clearly (no shame in that, it is a result, undoubtedly, which shows that further research should be carried out).

Finally, I would like to point out that, based on my expertise, I could not properly review the methods part on the climatic model, which I have not the expertise to revise properly. It seems interesting, though. One only thing that has to do with the matching of the model and the actual deformation values. Is that the best way to compare results? Why not showing the R2? It is fairly low, actually. From a visual perspective, there seems to be an acceptable agreement between the plots for the first two cases (figure 9a and 9b). In the third case (figure 9c), the model seems to oversmooth the peaks shown in the actual deformation trend. Is that only a climatological local effect, or it might also be a local lithological effect? The authors do not say anything on this, and I would have liked to read more discussion on the model unmatching.

Best regards.

Author Response

I have been asked to act as reviewer of the manuscript “Mapping active loess slopes in southwest area of Shanxi, China with Sentinel-1 InSAR analyses” by Xuguo Shi, Li Zhang, Yulong Zhong, Lu Zhang, Mingsheng Liao.

Most of my comments, even some general ones and suggestions, are contained as notes in the attached pdf. This letter only contains general comments.

The manuscript aims at presenting an approach to detect active deformations in slopes which lithology is exclusively consisting in loess, with particular focus on tryiing to identify precursors of failure in an event recorded in March 15th 2019. The authors also attempt to link simple climatological variables (precipitation, temperature) to the deformation pattern. In general, the idea proposed is interesting, despite not really novel. It is much a case study in a particular geological setting.

Despite the interesting idea, for how it is presented, the paper has several limitations and it can not yet be published as it is on the journal Remote Sensing unless moderate revisions are carried out to improve the publication and the work behind it.

Ans.: Thank you for your fruitful advices, we carefully revised our manuscript and corrected problems as much as possible.

The English language is, in general, good, even though some problems are present in a few expressions that the authors will find highlighted in the annotated pdf.

Ans.: Thank you for pointing this out, all the highlighted advices have been addressed.

On the terminology, I noticed some confusion between some terms that sould be used more consistenly to the scientific Literature. The very first definition given of loess is “a special landform formed by aeolian translport, loess accumulation and erosion ...”. The definition has two problem. The first is logical: the explanation contains the explained word. So it comes out that loess is formed by loess. The second is geological: loess is not a landform but a lithology, and specifically an aeolian deposit. Furhtermore, dealing with terminology, I would delete from the manuscript all the subjective judgements. So words such as special, tragedy, etc. They are not objective and should be avoided in academic English. The individual points were highlighted in the pdf.

Ans.: Thank you. We have revised all the confusing terminology and expressions.

The title could be chosen better. Slopes are not active or whatever. They are just slopes. Processes can be active, such as deformations or landslides, for example. Furthermore, I find not really suitable the use of the term “mapping”, whereas “detection” would be much more appropriate in this case. Detection and mapping are not synonyms. The detection has to do with the identification that a landslide is there. Mapping has to do with a formal delineation of a landslide that has been detected. This difference is substantial, and the use of these words as synonims leads to some consistency issues and, above all, makes the reader a promise that is not kept throughout the paper. I would suggest the title: “Detection (or identification) and characterization of active slope deformations with Sentinel-1 InSAR analyses in southwest area of Shanxi, China”.

Ans.: Thank you, we changed the title according to you suggestion as “Detection and characterization of active slope deformations with Sentinel-1 InSAR analyses in southwest area of Shanxi, China”.

The structure of the paper is acceptable in general. Only two main comments. The authors should focus more on highlighting adantages and limitations of the technique, future development, how they place this work in the state of the art, what is the research question they are answering, and to what extent the question can be said answered. What other question are open through this work.

Ans.: A paragraph has been added to address the state of art of InSAR in loess slope monitoring and the research question. The extent of solving the problem and opening questions have been listed in the discussion. (Line 58-68, 345-349 and 362-366)

A more conceptual issue for me has to do with the type of failure the authors try to find precursors for. The loss lithology, as an aeolian deposit, has a high isotropy and the grain size is quite homogeneous since the transport action is carried out by wind, which sorts the grains based on the current intensity. Similarly to other deposits that share these characteristics, this type of weak, unconsolidated and uncohesive lithologies show little to no precursors before failures, which makes them poorly suitable to monitoring. A confirmation of this is what you can see in the plots of Figure 7, where there is evidence of a very little deformation before the failure that caused the death of people. I would say that this approach is not suitable for setting up an early warning system, contrarily to what the authors state in the abstract and introduction. The best approach to keep low vulnerability to population is preventing them from living on the edge of escarpments of loess plateau, where the river incision, and scarp retreat inevitably pose a high risk to the population. For me the analyses show that this technique is not feasible, at least in this band. The authors should state this clearly (no shame in that, it is a result, undoubtedly, which shows that further research should be carried out).

Ans.: Thanks for your comments. Indeed, InSAR with long revisit times are definitely not suitable for landslide early warning. The failure of loess landslides might accelerate just a day or an hour before failure as Juang et al. (2019) stated. The temporal resolution of InSAR are not enough to achieve this. Actually, our result of the Xiangning landslide is just a trial to identify the possible cause of the failure since there is no ground measurements before the failure. Till now, there is still no official conclusion about the cause of landslide. Maybe our results can raise some discussions.

Juang, C. H.; Dijkstra, T.; Wasowski, J.; Meng, X., Loess geohazards research in China: Advances and challenges for mega engineering projects. Engineering Geology 2019, 251, 1-10.

Finally, I would like to point out that, based on my expertise, I could not properly review the methods part on the climatic model, which I have not the expertise to revise properly. It seems interesting, though. One only thing that has to do with the matching of the model and the actual deformation values. Is that the best way to compare results? Why not showing the R2? It is fairly low, actually. From a visual perspective, there seems to be an acceptable agreement between the plots for the first two cases (figure 9a and 9b). In the third case (figure 9c), the model seems to oversmooth the peaks shown in the actual deformation trend. Is that only a climatological local effect, or it might also be a local lithological effect? The authors do not say anything on this, and I would have liked to read more discussion on the model unmatching.

Ans.: To the best of our knowledge, comparison between modeled results and measurements are the best way. R2 have been calculated and shown in the text (Figure 10). The un-matching of P3 have been discussed (Line 310-320).

- Figure 1. What is this?

Ans.: This is the nine-dashed lines of China.

- Line 101, SRTM: You can add a reference (e.g. Farr & Kobrick, 2000 ).

Ans.: Agreed and revised (Line 114).

- Line 120, How did you unwrap the phase? Which algorithm? minimum cost flow, branch cut, other? Did you unwrap only spatially (like in 15) or even in time (24)?

Ans.: Agreed and revised. Three-dimensional phase unwrapping was used in our study (Line 132-133).

- Line 193, Replace ‘to avoid’ with ‘in case of’.

Ans.: Agreed and revised (Line 214).

- Line 225, In the manuscript you used mainly InSAR. Be consistent.

Ans.: Agreed and revised, we use InSAR in our paper.

- Line 230, ‘Those unstable slopes associated with mining activities are rendered in red.’.  The two mines in the insets (zoom) in figure 3 are in the red group? Looking at the figure the northernmost one seems to be in a valley not on a slope. I would associate this to a subsidence more then a slope instability. Did you plot the instabilities on a slope map? With only one looking geometry you are not able to discriminate from a simple subsidence or a slope movement. Furthermore, you showed two twelve days interferograms with a ground deformation of > 2 cm in different periods but, the deformation rate of the area is only 4 cm/yr. I would expect something larger (~30-40 cm/yr).

Since the a large displacement concentrates in a small area I would expect a loss of coherence in interferograms with temporal baseline longer then 24 days.

Ans.: Yes, we identify all the active slopes with a slope map. The problem you pointed out is right. Although the color range in Figure 4a is set as from -40 mm/yr to 40 mm/yr to better illustrate our results, there are a few MPs showing active deformations beyond this range. The most serious detected displacement rate in our study is over 140 mm/yr (Line 223-225). Besides, the displacements induced by mining activities are correlated with the evacuation rate. The fringes will gradually blur along with the evolution of evacuation. Unavoidable phase unwrapping errors will occur during the time series InSAR analysis process. Of course, the worst case is loss of coherence.

We compared our results from Sentinel-1 and from Envisat ASAR and ALOS PALSAR from Zhao et al. (2018) from 2007 to 2010. Although the temporal coverage is different, the retrieved displacement rates are very close. The maximum displacement rate reached over 114 mm/yr. Although the observation period is different, the observed displacement rates are in the same order of magnitude.

Zhao, C.; Liu, C.; Zhang, Q.; Lu, Z.; Yang, C., Deformation of Linfen-Yuncheng Basin (China) and its mechanisms revealed by Π-RATE InSAR technique. Remote Sensing of Environment 2018, 218, 221-230.

- Line 233, Replace ‘accumulative’ with ‘cumulative’.

Ans.: Agreed and revised (Line 253).

- Line 234, the cumulated deformation increase in time as expected. The size is almost the same in 20170908 (the first time in clearly visible with your colorbar) and in 20190314.

Ans.: Agreed and revised (Line 254-255).

- Line 246, ‘The results of DInSAR and time series InSAR analyses’  DinSAR ? not InSAR?

Ans.: Differential Interferograms were used in section 4.1, here we also use differential InSAR (Line 267).

- Line 247, ‘the official story’ If you write this you need to add a reference.

Ans.: Agreed and revised. A referred link was given in the reference (Line 269).

- Line 273 , Figure 8. ‘mean LOS displacement velocity maps overlaid on Google EarthTM maps.’ Did you try another colorscale? The red is saturated. I would expect to see different rates along the slope.

Ans.: Agreed and revised (Figure 9).

- Line 277 , ‘P2’, from the figure it seems that this point is on a flat area. It is a sinkhole, a subsidence area due to changes on the water level below the surface or something like this. I don't see a slope movement.

Ans.: It is an active slope, we redraw this figure as shown in Figure 8(b).

- Line 290 , Looking figures 4 and 5 P3 seems to be a "red" area this means that the deformation is associated to mine/anthropic activities. Could be this the explanation for the different behavior from the model?

Move the paragraph with the other explanation here.

Ans.: Agreed and revised (Line 310-320).

- Line 304, .Maybe is better to move it before. Move to Line 290.

Ans.: Agreed and revised (Line 313-316).

- Line 322, P2 seems to have an elliptical shape but you consider it as natural.

Ans.: The normal subsidence funnels induced by mining are of round or oval shape, which is very obvious in the interferograms. However, the shape are usually not round or oval as the mining activities goes on in the mean velocity map. Thus, we only considered elliptical shape presented in the differential interferograms in our study. P2 is the results from the time series analysis.

- Line 324, ‘In general, there might be precursory signals before slope failure events.’  Ref ?

Ans.: Agreed and revised. Two references have been included (Line 350).

Reviewer 4 Report

Authors used InSAR data to study the ground deformation in Shanxi (China). They also compared the deformation with the seasonal changes of temperatures and rain falls. They found a good agreement with their deformation model driven by climate variations with the non linear component of the displacement.The paper have some weakness.

You use only one looking geometry and so you are not able to discriminate between vertical or horizontal displacements. However you classify the observed signals in natural or anthropic slope instability even if some of the examples that you show seems to be located in flat areas. The majority of the observed signals could be related to simple subsidence that can produce a sinkhole but not the slope instability. I suggest to plot the instability areas on a slope map that will allow to better characterize the type. Another thing on the classification you used, circular and elliptical deformations could be also natural and at the same time you can have anthropic signal with other shapes. Another explanation could be very localized artifacts due to layover effect. In the area you have rapid topographic changes that could produce this small signals.

Another important concern I have is on the amplitude of the deformation that you show in the deformation map. You showed two interferogams with some areas that displace of >2cm in 12 days. Since the displacement is mainly linear I would expect a displacement rate up to 60 cm/yr while you obtain 4 cm/yr.

Please find other comments on the attached file. 

Author Response

(The authors gave the same response as above.)

Round 2

Reviewer 3 Report

I have no further comments for this paper. Many changes have been done, and now it seems more fluent and the ideas are better conveyed.

Best regards.